# Improved Boron Neutron Capture Therapy Using Integrin αvβ3-Targeted Long-Retention-Type Boron Carrier in a F98 Rat Glioma Model

**DOI:** 10.3390/biology12030377

**Published:** 2023-02-27

**Authors:** Kohei Tsujino, Hideki Kashiwagi, Kai Nishimura, Ryo Kayama, Kohei Yoshimura, Yusuke Fukuo, Hiroyuki Shiba, Ryo Hiramatsu, Naosuke Nonoguchi, Motomasa Furuse, Toshihiro Takami, Shin-Ichi Miyatake, Naonori Hu, Takushi Takata, Hiroki Tanaka, Minoru Suzuki, Shinji Kawabata, Hiroyuki Nakamura, Masahiko Wanibuchi

**Affiliations:** 1Department of Neurosurgery, Osaka Medical and Pharmaceutical University, Osaka 569-8686, Japan; 2Laboratory for Chemistry and Life Science, Institute of Innovative Research, Tokyo Institute of Technology, Yokohama 226-8503, Japan; 3Kansai BNCT Medical Center, Osaka Medical and Pharmaceutical University, Osaka 569-8686, Japan; 4Institute for Integrated Radiation and Nuclear Science, Kyoto University, Osaka 590-0494, Japan

**Keywords:** boron neutron capture therapy (BNCT), glioma, integrin, cyclic arginine-glycine-aspartate, biological target, albumin, drug delivery system

## Abstract

**Simple Summary:**

The development of a novel boron carrier is a key step in clinical boron neutron capture therapy (BNCT). We previously reported that maleimide-functionalized *closo*-dodecaborate albumin conjugate (MID-AC) with albumin as the drug delivery system is an effective boron carrier for the F98 glioma-bearing rat brain tumor model. In this study, the efficacy of BNCT with cRGD-MID-AC, a cyclic arginine-glycine-aspartate (cRGD) targeting integrin α_v_β_3_ added to MID-AC, was evaluated in a glioma-bearing rat brain tumor model. Although the cellular boron concentration of cRGD-MID-AC was lower than that of boronophenylalanine (BPA), in vitro neutron-irradiation experiments confirmed that the cell-killing effect of BNCT using cRGD-MID-AC was similar to that of BNCT using BPA. In vivo biodistribution showed a sufficient boron concentration in the tumor after intravenous administration. In neutron-irradiation experiments, the BNCT group using cRGD-MID-AC showed significantly prolonged survival compared to the untreated group, and long-term survivors were observed. This drug shows promise as a novel agent for BNCT.

**Abstract:**

Integrin α_v_β_3_ is more highly expressed in high-grade glioma cells than in normal tissues. In this study, a novel boron-10 carrier containing maleimide-functionalized *closo*-dodecaborate (MID), serum albumin as a drug delivery system, and cyclic arginine-glycine-aspartate (cRGD) that can target integrin α_v_β_3_ was developed. The efficacy of boron neutron capture therapy (BNCT) targeting integrin α_v_β_3_ in glioma cells in the brain of rats using a cRGD-functionalized MID-albumin conjugate (cRGD-MID-AC) was evaluated. F98 glioma cells exposed to boronophenylalanine (BPA), cRGD-MID-AC, and cRGD + MID were used for cellular uptake and neutron-irradiation experiments. An F98 glioma-bearing rat brain tumor model was used for biodistribution and neutron-irradiation experiments after BPA or cRGD-MID-AC administration. BNCT using cRGD-MID-AC had a sufficient cell-killing effect in vitro, similar to that with BNCT using BPA. In biodistribution experiments, cRGD-MID-AC accumulated in the brain tumor, with the highest boron concentration observed 8 h after administration. Significant differences were observed between the untreated group and BNCT using cRGD-MID-AC groups in the in vivo neutron-irradiation experiments through the log-rank test. Long-term survivors were observed only in BNCT using cRGD-MID-AC groups 8 h after intravenous administration. These findings suggest that BNCT with cRGD-MID-AC is highly selective against gliomas through a mechanism that is different from that of BNCT with BPA.

## 1. Introduction

Recently, boron neutron capture therapy (BNCT) has shown promise for the treatment of high-grade gliomas. BNCT is a particle radiation therapy that selectively destroys tumor cells through a nuclear reaction produced by thermal neutrons captured by boron-10 atoms. When glioma cells acquire boron-10 and are irradiated with low-energy thermal neutrons, the diameter of the high-linear-energy-transfer particles produced are limited to approximately 4–9 µm, and their path length is approximately equal to one tumor cell diameter (10 µm). Thus, the cell-killing effect of high-energy particle beams is theoretically limited to glioma cells that have acquired boron-10. Therefore, BNCT can target gliomas at the cellular level.

High-grade gliomas have residual glioma cells even after standard treatments, including chemoradiation following surgical resection, because they diffusely invade and infiltrate the normal brain parenchyma [1]. Thus, they can easily relapse from these residual glioma cells. The postoperative treatments need to selectively target such cells. High-grade gliomas require biologically targeted cell-selective therapy. BNCT may be suitable for the treatment of high-grade gliomas that grow invasively at the cellular level. In Japan, BNCT for head and neck cancer is now covered under insurance. In addition, clinical trials of reactor-based BNCT for high-grade gliomas have been conducted, and their efficacy has been reported [2,3,4,5,6,7]. BNCT using an accelerator-based neutron generator has also been shown to be effective against high-grade glioma [8]. Accelerator-based BNCT medical devices can be easily installed in any hospital and are expected to be widely used in clinical settings [8,9].

In past clinical trials of BNCT, the most commonly used boron carrier was boronophenylalanine (BPA). BPA is the only boron-10 carrier approved for use in BNCT by the Pharmaceuticals and Medical Devices Agency of Japan [8,9]. BPA is taken up by tumor cells via L-type amino acid transporter 1 (LAT1) [10]. However, there are some BPA-refractory conditions. This could be because, at the cellular level, some glioma cells may have difficulty taking up BPA, or BPA may be distributed sightly heterogeneously [11,12,13,14]. In clinical BNCT using BPA, higher-dose, long-term infusion, and continuous administration of BPA during irradiation have been used to increase its effectiveness [3,15]; however, long-retention-type boron carriers can achieve this with a single dose of a single agent.

Previously, we reported that maleimide-functionalized *closo*-dodecaborate albumin conjugate (MID-AC) is effective as a boron-10 carrier for BNCT in an F98 glioma-bearing rat brain tumor model [16]. MID-AC is the conjugation of MID and serum albumin and is known to be an effective drug delivery system (DDS) [16,17,18,19]. MID-AC is characterized by its long retention in tumor tissue. Serum albumin is used in the drug delivery mechanism to the tumor [16,19]. Although boron carriers have tumor tissue-accumulating properties, they have the disadvantage of not having a biological tumor target at the cellular level.

In the present study, integrin α_v_β_3_, which is overexpressed in many cancer cells, including high-grade gliomas [20,21,22,23], was bound to the boron carrier as a biological target system. Thus, cyclic RGD-functionalized *closo*-dodecaborate albumin conjugates with maleimide (cRGD-MID-AC) were developed by conjugating cyclic arginine-glycine-aspartate (cyclic RGD: cRGD), a known selective inhibitor of integrin α_v_β_3_ [24,25,26,27,28], and MID, which conjugates with albumin as well as MID-AC. In our previous study, cRGD-MID-AC showed significant tumor growth inhibition in a U87MG human glioma cell xenograft mouse model after neutron irradiation of subcutaneously implanted tumors [29]. Subcutaneous tumor models are limited in their ability to evaluate whether this novel boron-10 carrier can be applied in drug delivery for brain tumors. Thus, in this study, we evaluated the efficacy of BNCT using cRGD-MID-AC in an F98 glioma-bearing rat brain tumor model.

## 2. Materials and Methods

### 2.1. Boron Carriers

MID was synthesized according to the previously reported methods by Kikuchi et al. [19]. Albumin from lyophilized human serum powder (Sigma-Aldrich, Tokyo, Japan) was used. c[RGDfK(Mal)] was purchased from Synpeptide Co. Ltd. (Shanghai, China). MID, human serum albumin (HSA), and c[RGDfK(Mal)] were mixed in a ratio of 10:1:1 [29]. HSA contains Cys 34 and Lys residues. In the previous study, the fact was found that MID conjugates not only to Cys 34 but also to Lys residues in HSA [30]. Therefore, c[RGDfK(Mal)] was first introduced into Cys 34 in HSA, followed by of MID to Lys residues to prepare cRGD-conjugated boronate albumin [29]. The chemical structure and synthetic schema of cRGD-MID-AC are shown in Figure A1. BPA(L-isomer) was purchased from Interpharma Praha (Prague, Czech Republic) and was converted into a fructose complex [31]. All the boron-containing compounds were prepared as boron-10-enriched compounds.

### 2.2. Cell Culture

In the cellular uptake experiments, F98, C6 rat glioma, and 9L rat gliosarcoma cell lines were used. Each cell was provided by or purchased from as follows: F98 glioma cells, Dr. Rolf Barth (Ohio State University, Columbus, OH, USA); C6 glioma cells, the Japan Collection of Research Bioresources (JCRB) Cell Bank, National Institute of Biomedical Innovation (Osaka, Japan); 9L rat gliosarcoma cells, the American Type Culture Collection (ATCC; Manassas, VA, USA). All the cells were cultured in the medium that was used in our laboratory [16,32,33,34,35,36]. This medium contained Dulbecco’s modified Eagle’s medium (DMEM) and was supplemented with 10% fetal bovine serum, penicillin, streptomycin, and amphotericin B. All the materials for making this culture medium were purchased from Gibco Invitrogen (Grand Island, NY, USA). Of these cell lines, F98 glioma cells are particularly useful in evaluating therapeutic agents [16,32,33,34,35,36] because they simulate the behavior of human malignant gliomas in intracerebral implantation, including their high-infiltrative growth pattern and low immunogenicity [37,38]. The F98 glioma-bearing rat brain tumor model was used for in vivo evaluation.

### 2.3. F98 Glioma-Bearing Rat Brain Tumor Model

All animal experiments complied with “the Guide for the Care and Use of Laboratory Animals”, which was approved by both of the two facilities: the Animal Use Review Board and Ethical Committee of Osaka Medical and Pharmaceutical University (No. 21085-A) and the Institute for Integrated Radiation and Nuclear Science, Kyoto University (KURNS; Kumatori, Osaka, Japan) (No. 2021-18). In this study, 10-week-old male Fischer rats (Japan SLC, Shizuoka, Japan) were used. Their weight was approximately 200–250 g. They were anesthetized through an intraperitoneal injection of mixed anesthetics that are routinely used by our group [16,32,33,34,35,36]. Using a stereotactic frame (IMPACT-1000C connected to Legato 130, MUROMACHI KIKAI Co., Ltd., Tokyo, Japan), the rat’s head was fixed, and F98 glioma cells diluted in a 10 µL solution of DMEM with 1.4% agarose (Wako Pure Chemical Industries, Osaka, Japan) were implanted into the right brain. F98 glioma cells for therapeutic experiments (10^3^ cells) or biodistribution experiments (10^5^ cells) were implanted at a rate of 20 µL/min by an automated infusion pump into burr hole. The size of the burr hole was 1 mm, and it was made at 1 mm posterior to the bregma, 4 mm to the right lateral side. These surgical procedures are routinely used by our group [16,32,33,34,35,36].

### 2.4. In Vitro Cellular Uptake Experiments

For the cellular uptake of boron, F98, C6 glioma, and 9L gliosarcoma cells were used. Cells (5 × 10^5^) were seeded and incubated in a 100 mm dish (Becton, Dickinson, and Company, Franklin Lakes, NJ, USA) with the culture medium for 4 days. Just before the cells in the dish became confluent, the medium was exchanged for a culture medium with 10 µg B/mL BPA, cRGD-MID-AC, or cRGD + MID (MID and cRGD were simply mixed at a ratio of 10:1), and the cells were incubated for an additional 1, 6, and 24 h at 37 °C. In experiments to measure the retention rate of boron, a cell culture medium with 10 µg B/mL of BPA, cRGD-MID-AC, or cRGD + MID was incubated for 24 h. Then, the medium was exchanged for a boron-free medium and incubated for 1, 6, or 24 h. The medium with the boron carriers or the boron-free medium was removed, and the cells were washed twice with 4% phosphate-buffered saline (PBS), detached by trypsin-ethylenediamine tetraacetic acid solution, and all the cells in the dish were collected. The culture medium was then added in the dish, and the cells were counted after centrifugation twice (at 200× *g* for 5 min). The cells were then dissolved overnight in a 1 N nitric acid solution (Wako Pure Chemical Industries, Osaka, Japan), and the amount of intracellular boron was measured by inductively coupled plasma atomic emission spectroscopy (ICP-AES; iCAP6300 emission spectrometer, Hitachi, Tokyo, Japan). The intracellular boron concentrations were defined as µg boron (B)/10^9^ cells.

### 2.5. In Vitro Neutron-Irradiation Experiments

The cytotoxicity of each boron carrier in BNCT was evaluated using colony-forming assays. F98 glioma cells were used for the in vitro neutron-irradiation experiments. They were incubated in 150 cm^2^ tissue culture flasks with 20 mL of each medium for four groups: group 1, neutron only (boron-free medium); group 2, BPA; group 3, cRGD-MID-AC; and group 4, cRGD + MID. All boron-exposure groups were exposed for 2.5 h to 10 µg B/mL of boron-containing medium derived from each drug. Neutron irradiation was then performed. Cells were irradiated for 10, 20, and 30 min at a reactor power of 1 MW with a neutron flux of 1.1 × 10^9^ neutrons/cm^2^/s at KURNS. After neutron irradiation, cells from each sample were collected and counted. The cell solution was diluted to the predetermined number of cells, and the cells were seeded in a 60 mm dish (Becton, Dickson, and Company, Franklin Lakes, NJ, USA) (Three dishes were prepared per group and per number of cells). The cells were then incubated for seven days. Finally, they were fixed with 90% ethanol and stained with Giemsa. The survival fraction (SF) was calculated by counting the number of colonies consisting of more than 50 cells and dividing it by the number of colonies of the control group. In this experiment, the RBE (relative biological effectiveness) of the neutron beam and the CBE (compound biological effectiveness) of each boron carrier were estimated by considering the linear-quadratic (LQ) model obtained from the X-ray irradiation of F98 glioma cells and the physical dose to achieve SF = 0.1 in each group [39].

### 2.6. Biodistribution of Boron in the F98 Glioma-Bearing Rats after Intravenous Administration of Each Boron Carrier

Approximately 12–14 days after implantation of 10^5^ F98 glioma cells, when the tumor was expected to have grown sufficiently, each boron carrier was administered at 12 mg boron (B)/kg body weight (b.w.). At each predetermined time, the rats were euthanized, and each tissue (the tumor, brain, blood, heart, lung, liver, kidney, spleen, skin, and muscle samples) was removed. Each organ was weighed and digested with 1 N nitric acid solution after weighing. The amount of boron in each organ was measured by ICP-AES. All results (boron concentrations) were defined as µg boron (B)/g.

### 2.7. Survival Analysis of the In Vivo Neutron-Irradiation Experiments

Fifteen days after the implantation of 10^3^ F98 glioma cells, a total of 33 F98 glioma-bearing rats were randomly divided into the following six groups: group 1, untreated control group (untreated); group 2, neutron-irradiated control group (neutron only); group 3, neutron irradiation after 2.5 h of BPA intravenous administration (i.v.) (BNCT using BPA 2.5 h); group 4, neutron irradiation after 8 h of BPA i.v. (BNCT using BPA 8 h); group 5, neutron irradiation after 2.5 h of cRGD-MID-AC i.v. (BNCT using cRGD-MID-AC 2.5 h); and group 6, neutron irradiation after 8 h of cRGD-MID-AC i.v. (BNCT using cRGD-MID-AC 8 h).

All rats were anesthetized through an intraperitoneal injection of the anesthetics, and BPA or cRGD-MID-AC was administered to the assigned experimental groups. Only their heads were irradiated with neutrons at the KURNS. They were irradiated for 20 min at a reactor power of 5 MW and a neutron flux of 9.6 × 10^8^ neutrons/cm^2^/s at the Heavy Water Irradiation Facility at KURNS. All rats were observed until death or euthanasia. In addition, the therapeutic effects were evaluated by Kaplan–Meier survival curves, and the percent increase in life span (%ILS) was calculated by the following equation: (the median survival times; MST of each BNCT group–MST of untreated group) × 100/(MST of untreated).

### 2.8. Estimated Physical Dose and Biologically Photon-Equivalent Dose

The physical dose was determined using the dose calculated from the thermal, epithermal, fast neutron, and gamma rays of the irradiated neutrons. The equation D_B_ + D_N_ + D_H_ + D_γ_ was used, and each factor corresponded to ^10^B(n,α)^7^Li, ^14^N(n,p)^14^C, and ^1^H(n,n)^1^H capture reactions and γ-rays, respectively. What D_B_, D_N_, D_H_, and D_γ_ mean and how to calculate them have already reported, and the physical doses to the brain and brain tumors were calculated for each group and calculated according to [32,36]. The estimated photon-equivalent dose was calculated by the following equation: D_B_ × CBE + D_N_ × relative biological effectiveness (RBE_N_) + D_H_ × relative biological effectiveness (RBE_H_) + D_γ_. Both the physical doses and estimated photon-equivalent doses were corrected according to the previous report [32,36]. CBE refers to the biological effectiveness ratio in boron neutron capture reaction that is specific to the irradiated tissue or boron carrier. In the previous report, the CBE factor for normal brain was defined as 1.35 in BNCT with BPA [2,40]. In order to calculate the estimated photon-equivalent doses for in vivo irradiation experiment, the CBE for BPA and cRGD-MID-AC was calculated by the results obtained in the in vitro experiment.

### 2.9. Statistical Analysis

In the in vitro cellular uptake experiments, the intracellular boron concentrations in all cell lines were evaluated by the Student’s *t*-test. Survival times were evaluated by Kaplan–Meier curves. Log-rank tests were used to determine significant differences between the groups. For all tests, statistical *p*-values of less than 0.05 were evaluated as a significant difference. All the results were analyzed using JMP^®^ Pro version 15.1.0. software (SAS, Cary, NC, USA).

## 3. Results

### 3.1. In Vitro Cellular Boron Uptake Experiments in F98, C6 Glioma, and 9L Gliosarcoma Cells

The boron concentrations in F98, C6 glioma, and 9L gliosarcoma cells after exposure to each boron carrier from 1 h to 24 h are shown in Figure 1A–C. The retention rates of boron for BPA, cRGD-MID-AC, and cRGD + MID are shown in Figure 1D–F. The cellular boron concentrations of each cell at 1, 6, and 24 h after exposure to 10 µg B/mL of each boron carrier and at 1 (24 + 1), 6 (24 + 6), and 24 (24 + 24) h after 1, 6, and 24 h of additional incubation, when the medium was changed to boron-free, are shown in Table A1, Table A2 and Table A3. The boron concentrations of BPA and cRGD-MID-AC from 1 to 24 h in all cells gradually increased over time. In contrast, the boron concentrations of cRGD + MID were the highest at 6 h and did not show significant differences until 24 h later in all cells (Figure 1A–C). The boron concentration of BPA from 1 h to 24 h in all cell lines was significantly higher than that of cRGD-MID-AC and cRGD + MID (*p* < 0.05). However, from 24 + 1 h to 24 + 24 h, the boron concentration of cRGD-MID-AC in all cell lines was significantly higher than that of BPA (*p* < 0.05, except 9L 24 + 24 h). The retention rates of boron for BPA, cRGD-MID-AC, and cRGD + MID after 1 h of additional incubation when the medium was changed to boron-free were 14.7%, 74.6%, and 65.2% in F98 glioma cells; 15.7%, 88.8%, and 78.4% in C6 glioma cells; and 23.3%, 85.6%, and 59.1% in 9L rat gliosarcoma cells, respectively (Figure 1D–F). After 6 and 24 h of additional incubation, the retention rates of boron for BPA, cRGD-MID-AC, and cRGD + MID were 12.0%, 72.3%, and 56.4% (6 h) and 9.5%, 34.3%, and 21.1% (24 h) in F98 glioma cells; 15.6%, 82.8%, and 75.3% (6 h) and 7.4%, 26.5%, and 22.6%(24 h) in C6 glioma cells; and 16.3%, 70.7%, and 54.3% (6 h) and 12.7%, 31.3%, and 24.7% (24 h) in 9L rat gliosarcoma cells, respectively (Figure 1D–F). The concentration of boron retained by tumor cells in each group decreased rapidly in BPA after replacement with boron-free medium, whereas it decreased gradually over time in all the other groups.

### 3.2. Neutron Irradiation in the In Vitro Experiments

The results of the in vitro neutron-irradiation experiments are shown in Figure 2. The physical dose of each neutron irradiation was 0 Gy (0 min), 0.48 Gy (10 min), 1.01 Gy (20 min), and 1.21 Gy (30 min). The SF of BPA and cRGD-MID-AC in the 10 min irradiation group was significantly different (BPA: 0.09, cRGD-MID-AC: 0.35, *p* < 0.001); however, the SF of BPA and cRGD-MID-AC in the 20 and 30 min irradiation groups was not significantly different (20 min: BPA; 0.05, cRGD-MID-AC; 0.04, *p* = 0.72, 30 min: BPA; 0.04, cRGD-MID-AC; 0.10, *p* = 0.08). Using the linear-quadratic (LQ) model obtained from X-ray irradiation of F98 glioma cells, the doses to calculate the CBE factors for the BPA, cRGD-MID-AC, and cRGD + MID groups were estimated from the colony-forming assays. The physical doses required to achieve SF = 0.1 in the BPA, cRGD-MID-AC, and cRGD + MID groups were 0.75, 0.85, and 1.48 Gy, respectively. (In the LQ model, the corresponding dose was 6.45 Gy [36].) The CBE factors of the BPA, cRGD-MID-AC, and cRGD + MID groups (determined by the calculated RBE) were 2.69, 2.26, and 0.75, respectively.

### 3.3. Biodistribution of Boron in the F98 Glioma-Bearing Rats after Intravenous Administration of Each Boron Carrier

Boron concentrations in each organ were evaluated at 2.5, 8, and 24 h after intravenous administration of cRGD-MID-AC or BPA. In the case of cRGD-MID-AC, the boron concentrations in the tumor were 10.1 ± 1.6 (0.8 ± 0.2 in the brain and 41.6 ± 5.6 in the blood), 17.0 ± 1.8 (0.9 ± 0.1 in the brain and 40.3 ± 8.4 in the blood), and 13.1 ± 1.9 (0.7± 0.1 in the brain and 17.7 ± 2.3 in the blood) µg B/g at 2.5, 8, and 24 h, respectively. The boron concentration in the tumor was the highest at 8 h after intravenous administration of cRGD-MID-AC and tended to be retained for a longer period of 24 h. In contrast, in the case of BPA, the boron concentrations were 20.6 ± 2.2 (5.5 ± 0.6 in the brain and 7.7 ± 0.5 in the blood), 18.2 ± 2.9 (5.3 ± 0.5 in the brain and 4.8 ± 0.3 in the blood), and 8.2 ± 0.8 (2.3 ± 0.3 in the brain and 2.9 ± 0.4 in the blood) µg B/g at 2.5, 8, and 24 h, respectively [16]. The boron concentrations in the tumor were the highest 2.5 h after intravenous administration of BPA and decreased gradually. In addition, after every hour, the tumor/normal brain ratio of cRGD-MID-AC was much higher than that of BPA. Table 1 presents a summary of our results, and Figure 3 shows the boron concentrations in each organ and the ratio of tumor to normal brain tissue (T/Br ratio).

### 3.4. Survival Analysis of the In Vivo Neutron-Irradiation Experiments

Neutron-irradiation experiments were performed at both 2.5 and 8 h because the boron concentration in the tumor was the highest at 2.5 h for BPA and at 8 h for cRGD-MID-AC according to the biodistribution experiments. The treatment effect was evaluated using Kaplan–Meier curves (Figure 4A,B). Each MST and %ILS value is shown in Table 2.

Statistically significant differences were observed between the untreated group and all BNCT groups evaluated by the log-rank test (Table 2). Ninety days after the F98 glioma cell implantation, one F98 glioma-bearing rat survived for a long time only in the group of BNCT using cRGD-MID-AC 8 h.

### 3.5. Estimation of Physical and Biologically Photon-Equivalent Doses

The physical and photon-equivalent doses for the brain and tumor by neutron-irradiation experiments were calculated using the CBE factor of each boron carrier obtained as a reference and in vitro neutron-irradiation experiment and the mean boron concentrations in the tumor obtained from in vivo biodistribution experiments. RBE_N_ and RBE_H_ were adopted as 3.0 according to the previous report [41]. The calculated photon-equivalent doses for the brain tumor obtained with BNCT using BPA at 2.5 h and 8 h were 10.9 Gy-Eq and 9.9 Gy-Eq, respectively, and with BNCT using cRGD-MID-AC at 2.5 h and 8 h were 5.8 Gy-Eq and 9.5 Gy-Eq, respectively (Table 3).

## 4. Discussion

cRGD-MID-AC was adapted for BNCT against an experimental brain tumor model for the first time. A significant difference between untreated and BNCT using cRGD-MID-AC was observed by log-rank test, which suggested that BNCT using cRGD-MID-AC is effective in the experimental brain tumor model (Figure 4A,B, and Table 2). In addition, long-term survivors were observed in BNCT using cRGD-MID-AC for 8 h, whereas no long-term-surviving individuals were observed in the other groups. This result suggested that cRGD-MID-AC accumulates in glioma cells with high integrin expression. In other words, the effect of cRGD-MID-AC may have been pronounced in the F98 glioma-bearing rats with high integrin expression [42].

Although MID-AC accumulates in tumor cells and can prolong the boron concentration in the tumor for up to 24 h after intravenous administration, the boron concentration delivered to the brain tumor by MID-AC was still low (<8.5 µg B/g) [16]. Thus, MID-AC has been modified by conjugation with cRGD, which binds strongly to integrins, especially integrin α_v_β_3_, and this conjugate was thought to contribute to improving the efficacy of BNCT against gliomas. Integrins are a family of cell–cell and cell–extracellular matrix adhesion molecules [43]. Among them, integrin α_v_β_3_ is overexpressed in glioma cells, whereas its expression in normal cells is low and is related to invasion, proliferation, and angiogenesis of tumor cells [20,21,22,23,43,44,45]. The CENTRIC study (phase III trial), which investigated the efficacy of cilengitide, a selective integrin α_v_β_3_ inhibitor, as antitumor therapy against glioblastoma in combination with standard postoperative chemoradiation therapy, reported that cilengitide did not show improved outcomes, and neither progression-free survival nor overall survival was significantly prolonged [46]. However, the therapeutic approach of targeting integrin remains a sound strategy for high-grade gliomas, and there have been various studies on treatment using RGD [28,46]. In addition, the CENTRIC study reported no additional toxic effects of cilengitide. In the case of high-grade gliomas, higher the malignancy of the glioma, the higher the expression of integrin α_v_β_3_, which is a poor prognostic factor. Therefore, this novel boron carrier has the potential to be effective for high-grade gliomas. In this study, cRGD was used as a tumor target domain for binding with MID-AC.

Thus, cRGD-MID-AC, which has human serum albumin as a DDS and integrin α_v_β_3_ as a tumor-targeting system, has been developed. The efficacy of human serum albumin and its accumulation in the tumor by conjugation with albumin have been reported in previous studies [16,17,18,19,47,48]. In this experiment, immunogenic HSA was used instead of rat serum albumin. When considering this albumin-containing drug as a pre-clinical study, the results of this study require to be interpreted carefully, taking into account the effects of immunogenicity in a rat brain tumor model. The most significant feature of cRGD-MID-AC is the high retention of cellular boron concentration owing to the tumor accumulation mechanism of albumin as well as MID-AC. In vitro cellular uptake experiments showed that the cellular boron concentration of BPA was highest at all times from 1 h to 24 h. The cellular boron concentration of cRGD-MID-AC increased gradually over time, and the retention rate after 24 h of exposure was much higher than that of BPA (Figure 1). As BPA is rapidly cleared after exposure, it must be continuously administered intravenously during neutron irradiation in clinical BNCT [3,15]. In contrast, cRGD-MID-AC can retain cellular boron for a long time after the completion of exposure, which is a favorable aspect for BNCT. Furthermore, in assessing the novel boron carrier in BNCT, it is necessary to evaluate the biological effects of BNCT, which are shown by the CBE factor specific for each boron carrier. In this study, based on in vitro neutron-irradiation experiments, the CBE for cRGD-MID-AC was calculated as 2.26 (CBE to be 2.69), which was found to have a sufficient cell-killing effect in BNCT as compared to that of BPA even though the cellular boron concentration of cRGD-MID-AC was lower than that of BPA (Figure 2).

The efficacy of cRGD-MID-AC as a tumor-targeting system was demonstrated by in vivo biodistribution experiments (Figure 3). In the case of cRGD-MID-AC, the boron concentration in the tumor was as high as that in BPA. Notably, the distribution of boron-10 in normal brain tissue was much lower with cRGD-MID-AC than with BPA, indicating an even lower level of damage from neutron irradiation. Thus, cRGD-MID-AC may provide boron-10 more selectively than BPA, resulting in a safer or more intense boron neutron capture reaction with a longer irradiation time for high-grade gliomas. Furthermore, a previous study showed that BNCT using cRGD-MID-AC is much more effective than BNCT using MID-AC in a U87MG xenograft subcutaneous tumor mouse model because of conjugation with RGD and might enhance therapeutic efficacy against high-grade gliomas with high integrin expression [29]. In addition, compared with the intravenous administration of 20 mg B/kg of MID-AC to the F98 glioma-bearing rat brain tumor model in our previous study, the boron concentration in the tumor was higher when 12 mg/kg of cRGD-MID-AC was administered intravenously [16]. These results suggest that the bonding of cRGD to MID-AC, as a tumor-targeting system, is expected to improve therapeutic efficiency.

In vivo neutron-irradiation experiments showed the efficacy of BNCT using cRGD-MID-AC (Figure 4A,B). No significant differences were observed between the BNCT with BPA and BNCT with cRGD-MID-AC. Concerning long-term survival in the cRGD-MID-AC 8 h group, which was not observed in the BNCT with BPA group, it is possible that prolonged exposure may enhance cellular boron accumulation in the tumor (even in vitro), and this phenomenon was confirmed in the survival evaluation after BNCT. BPA has been found to be widely distributed in the cytoplasm and cell nucleus. However, the cellular distribution of cRGD-MID-AC remains unclear. Although the interaction between RGD and integrins has long been known, the mechanism of internalization by binding of RGD peptides to integrins has not yet been elucidated. However, it has been reported that at least the endocytosis of integrin α_v_β_3_ is mediated via clathrin-dependent endocytosis or uncoated vesicles [49,50]. Schraa et al. showed that internalization of monomeric RGD ligands is independent of their α_v_β_3_ receptor and occurs via a liquid-phase endocytic pathway. In contrast, multimeric RGD molecules are co-internalized with their receptor, with evidence supporting the aggregation and clustering of integrins [51]. More recently, Sancey et al. reported a peptide-like scaffold with four cRGD motifs, called RAFT-RGD, that target integrin α_v_β_3_ and promote integrin cluster formation. They have further demonstrated that the addition of 1 μmol/L of this molecule (RAFT-RGD) increases the internalization of α_v_β_3_ via clathrin-coated vesicles by 79% [52]. Our study did not examine the internalization rate of integrin α_v_β_3_ or the extent to which our DDS applied with cRGD contributed to the internalization of integrin α_v_β_3_. However, we will explore these in future studies to better understand and improve the potential of DDS to specifically deliver and retain boron in target cells.

The accumulation of cRGD-MID-AC in glioma cells in the brain partly results from the disruption of the blood–brain barrier (BBB). In BNCT for high-grade gliomas, the requirements for boron-10 carriers include low intrinsic toxicity, high boron accumulation for target lesions, low uptake into normal tissues, water solubility [35,53], and the ability to overcome the BBB and the blood–brain tumor barrier (BBTB) [54,55,56]. Currently, only two boron carriers, namely BPA and sodium borocaptate (BSH), are used in clinical BNCT. BSH contains 12 boron atoms per molecule, reaches tumor cells through disruption of the BBB, and is not cell selective [53]. Because MID, which constitutes cRGD-MID-AC, is a derivative of BSH, cRGD-MID-AC, like BSH, is thought to reach tumor cells by disrupting the BBB. In addition, because integrin α_v_β_3_ is highly expressed in the neovascular vessels surrounding tumors [22,23], cRGD-MID-AC may accumulate more in these neovascular vessels than in normal vessels. If cRGD-MID-AC crosses the BBB, the accumulation of boron in the normal brain is expected to be slightly higher; however, the fact that it is much lower than that of BPA suggests that it does not pass through the healthy, intact BBB. cRGD-MID-AC and MID-AC contain albumin as the DDS, and the accumulation of sufficient boron concentration in tumor cells may result from the albumin conjugate in addition to the MID properties. Therefore, it is very appealing that cRGD-MID-AC with 12 boron atoms per molecule can reach glioma cells and remain in these cells for a long time owing to the properties of human serum albumin.

In another perspective study, cRGD-MID-AC was shown to have potential clinical applications. Recently, in neuro-oncology, positron emission tomography (PET) imaging has attracted much attention because it allows noninvasive assessment of molecular and metabolic processes [57]. In patients with brain tumors, particularly gliomas, it is important to assess treatment-related changes such as radiation necrosis and indications for targeted therapies such as integrins [57]. It has been shown that ^18^F-Galacto-RGD PET can assess integrin α_v_β_3_ expression in mouse tumor models and in patients with head and neck cancer. A correlation has also been demonstrated between galacto-RGD uptake in patients with malignant gliomas and the expression of integrin α_v_β_3_ in the corresponding tumor samples [58,59]. These findings suggest that imaging of integrin α_v_β_3_ expression in patients with malignant glioma by ^18^F-Galacto-RGD positron emission tomography has already been established [23]. New RGD peptide ligands are also being developed for PET imaging of α_v_β_3_ integrin, which may become even more important as integrin-related therapeutic decisions and treatments become relevant for patients with glioma [60]. Therefore, the application of these techniques would be useful for visually identifying eligible cases for integrin-targeted BNCT. In this study, only rat gliomas and gliosarcoma cells were evaluated. For cRGD-MID-AC to be clinically applicable, it will also be necessary to evaluate other more heterogeneous tumor models in which therapeutic effects can be expected depending on the degree of integrin α_v_β_3_ expression.

Only a few boron-10 carriers can be as effective as or even more effective than BNCT with BPA against high-grade gliomas, especially when administered intravenously [16,34]. This study showed that BNCT with cRGD-MID-AC could precisely target glioma cells even in the brain. This biological targeting of F98 glioma cells that overexpress integrins α_v_β_3_, apart from LAT-1-targeted BPA, would provide a novel target for BNCT against high-grade gliomas with a heterogeneous nature. Because cRGD-MID-AC has a very low boron distribution in normal brain tissues, it would be expected to further improve the therapeutic effect owing to the different biological targets from to that of BPA.

## 5. Conclusions

cRGD-MID-AC, cyclic RGD-functionalized *closo*-dodecaborate albumin conjugates with maleimide, has been shown to have a therapeutic effect in BNCT in an experimental F98 glioma-bearing rat brain tumor model. The effect of albumin as a DDS increases the blood residence time, and the effect of cRGD as a biological target increases tumor selectivity, which provides more intensive BNCT against high-grade gliomas compared with that with BPA. The targeting strategy of cRGD can achieve a higher tumor boron concentration than the original long-retention type boron carrier, MID-AC, and can improve the therapeutic intensity of BNCT. Furthermore, cRGD-MID-AC is appealing because it can be administered intravenously like BPA. As in the case of MID-AC, it is expected to contribute to the flexible application of neutron irradiation and the performance of BNCT by retaining boron in the tumor for a long period of time. In addition, because of different accumulation mechanisms, it may be possible to use it in combination with BPA-based BNCT to make it a multi-targeted NCT.

## Figures and Tables

**Figure 1 biology-12-00377-f001:**
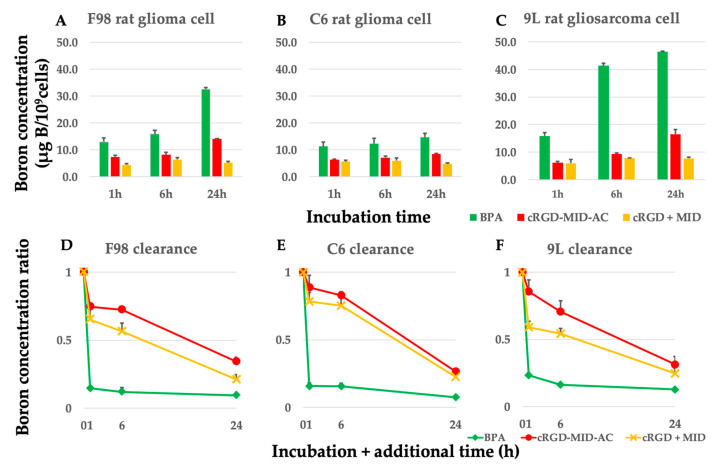
(**A**–**C**) Boron concentrations in all cell lines. All cell lines were incubated with 10 µg B/mL of BPA, cRGD-MID-AC, and cRGD + MID for 1, 6, and 24 h. The bar in each result indicates the standard deviation. (**D**–**F**) To show the retention ratio of boron of BPA, cRGD-MID-AC, and cRGD + MID, each cell line was incubated with 10 µg B/mL of BPA, cRGD-MID-AC, and cRGD + MID for 24 h with additional incubation with a boron-free medium for 0, 1, 6, and 24 h. The bar in each result indicates the standard deviation. The boron concentration of BPA from 1 h to 24 h in all cell lines was significantly higher than that of cRGD-MID-AC and cRGD + MID (*p* < 0.05).

**Figure 2 biology-12-00377-f002:**
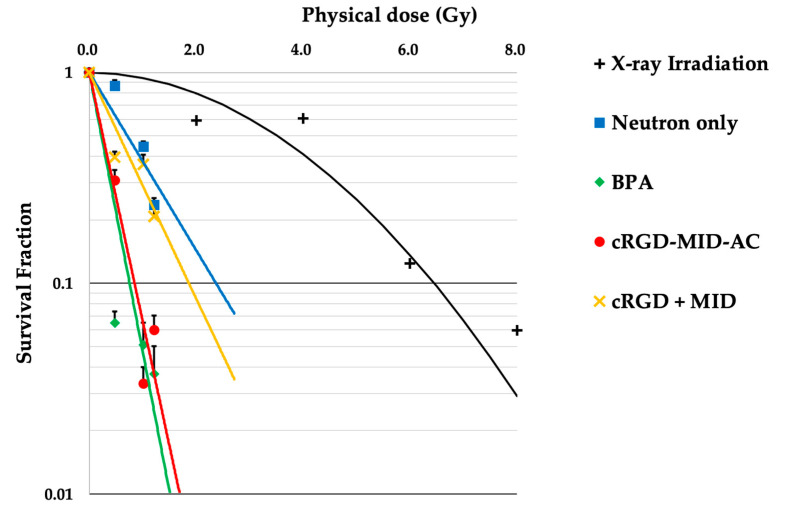
This figure shows the linear-quadratic (LQ) model for F98 glioma cells after X-ray irradiation and survival lines after neutron irradiation. The survival fractions corresponding to each physical dose were plotted, and the approximate line was drawn for each group. The error bars indicate standard error (SE).

**Figure 3 biology-12-00377-f003:**
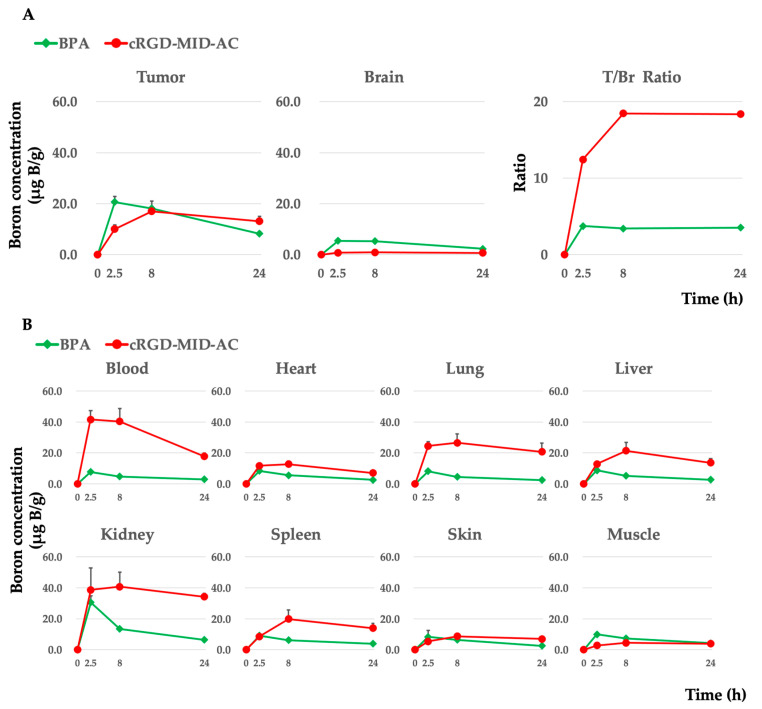
(**A**) Boron concentrations in the tumor and brain (normal brain) in the F98 glioma-bearing rat brain tumor model after the implantation of 10^5^ F98 glioma cells. Each boron carrier was administered with 12 mg boron (B)/kg body weight (b.w.) doses. At 2.5, 8, and 24 h after intravenous administration, each rat was euthanized, and each tissue was removed. The boron concentration in each tissue was measured by ICP-AES. The mean boron concentration of each tissue was defined as μg B/gram ± standard deviation (SD). In addition, the tumor/brain boron concentration ratio (T/Br ratio) is shown. (**B**) Boron concentrations in each tissue except the tumor and the brain are shown.

**Figure 4 biology-12-00377-f004:**
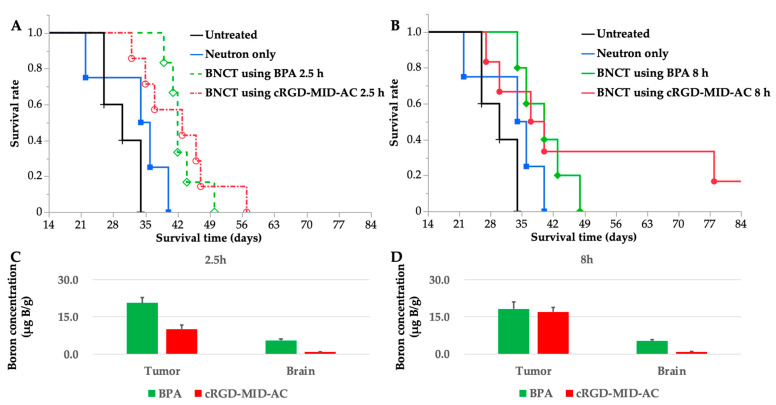
(**A**,**B**) These figures show the Kaplan–Meier survival curves for F98 glioma-bearing rats after neutron irradiation. Survival times (days) after implantation of F98 glioma cells are plotted. Six groups were created; untreated, neutron only, BNCT using boronophenylalanine (BPA) 2.5 h, BNCT using BPA 8 h, BNCT using cyclic RGD-functionalized *closo*-dodecaborate albumin conjugates with maleimide (cRGD-MID-AC) 2.5 h, and BNCT using cRGD-MID-AC 8 h. (**A**) Kaplan–Meier survival curves for untreated, neutron only, BNCT using BPA 2.5 h, and BNCT using cRGD-MID-AC 2.5 h. (**B**) Kaplan–Meier survival curves for untreated, neutron only, BNCT using BPA 8 h, and BNCT using cRGD-MID-AC 8 h. In the case of BNCT using cRGD-MID-AC 8 h, one F98 glioma-bearing rat survived for a long time: 90 days after the F98 glioma cell implantation, when the other animals had died or were euthanized. (**C**) Boron concentrations of tumor and brain (normal brain) at 2.5 h after intravenous administration of BPA or cRGD-MID-AC are shown. (**D**) Boron concentrations of tumor and brain (normal brain) at 8 h after intravenous administration of BPA or cRGD-MID-AC are shown. All the error bars indicate standard deviation (SD).

**Table 1 biology-12-00377-t001:** Boron concentrations in the tumor, brain (normal brain), and blood after intravenous administration of each boron carrier in F98 glioma-bearing rats.

Boron Carrier ^a^	Time ^b^ (h)	n ^c^	Boron Concentration ± SD (µg B/g) ^d^			Ratio	
			Tumor	Brain	Blood	T/Br ^e^	T/Bl ^f^
cRGD-MID-AC	2.5	4	10.1 ± 1.6	0.8 ± 0.2	41.6 ± 5.6	12.5	0.2
	8	4	17.0 ± 1.8	0.9 ± 0.1	40.3 ± 8.4	18.5	0.4
	24	4	13.1 ± 1.9	0.7 ± 0.1	17.7 ± 2.3	18.4	0.7
BPA	2.5	4	20.6 ± 2.2	5.5 ± 0.6	7.7 ± 0.5	3.8	2.7
	8	3	18.2 ± 2.9	5.3 ± 0.5	4.8 ± 0.3	3.4	3.8
	24	4	8.2 ± 0.8	2.3 ± 0.3	2.9 ± 0.4	3.6	2.8

^a^ Each boron carrier was administered at 12 mg/kg body weight. ^b^ Time indicates the period from the time of intravenous administration of each boron carrier to the time of euthanasia of F98 glioma-bearing rats. ^c^ n indicates number of Fischer rats. ^d^ Boron concentration ± SD (µg B/g) is the value measured by ICP-AES for the acquired boron concentration of each organ listed in the table and is expressed as the mean boron value (µg B/g: weight of organ) ± standard deviation. ^e^ T/Br indicates the tumor-to-brain ratio. ^f^ T/Bl indicates the tumor-to-blood ratio.

**Table 2 biology-12-00377-t002:** Survival times of F98 glioma-bearing rats after neutron irradiation.

Group	n ^a^	Survival Time (Days)			%ILS ^c^	*p*-Value ^d^
		Mean ± SD	Median	95% CI ^b^		
Untreated	5	30.0 ± 4.0	30.0	26–34	-	-
Neutron only	4	33.0 ± 7.7	35.0	22–40	16.7	0.18
BNCT using BPA 2.5 h	6	43.0 ± 3.8	42.0	39–50	40.0	0.0011
BNCT using BPA 8 h	5	40.2 ± 5.6	40.0	34–48	33.3	0.0079
BNCT using cRGD-MID-AC 2.5 h	7	42.4 ± 8.6	43.0	32–47	43.3	0.0033
BNCT using cRGD-MID-AC 8 h	6	50.3 ± 26.8	38.5	27-	28.3	0.0499

^a^ n indicates number of Fischer rats per group. ^b^ CI is confidence interval. ^c^ Percent increase in life span (%ILS) was defined relative to the mean survival time (MST) of the untreated group. ^d^ *p*-values were calculated using the log-rank test compared to untreated based on the results obtained from the Kaplan–Meier curves in the neutron-irradiation experiment for F98 glioma-bearing rats.

**Table 3 biology-12-00377-t003:** Physical and estimated photon-equivalent doses for brain or tumor in F98 glioma-bearing rats.

Group	Physical Dose ^a^ (Gy)		Photon-Equivalent Dose ^b^ (Gy-Eq)	
	Brain	Tumor	Brain	Tumor
Untreated	0.0	0.0	0.0	0.0
Neutron only	1.6	1.6	2.6	2.6
BNCT using BPA 2.5 h	2.2	4.6	3.4	10.9
BNCT using BPA 8 h	2.2	4.2	3.3	9.9
BNCT using cRGD-MID-AC 2.5 h	1.5	2.9	- *	5.8
BNCT using cRGD-MID-AC 8 h	1.8	4.6	- *	9.5

^a^ The physical dose is attributed to the ^10^B(n,α)^7^Li, ^14^N(n,p)^14^C, and ^1^H(n,n)^1^H reactions produced by thermal, epithermal, and fast neutron fluxes and gamma rays in the irradiation neutrons. It is calculated using the following equation: physical radiation dose (Gy) = D_B_ + D_N_ + D_H_ + D_γ_. ^b^ The estimated photon-equivalent dose was calculated using the following equation: D_B_ × compound biological effectiveness + DN × relative biological effect of nitrogen (RBE_N_) + D_H_ × relative biological effect of hydrogen (RBE_H_) + D_γ_. RBE_N_ and RBE_H_ are 3.0. In the case of BPA, the CBE factor for the normal brain tissue was 1.35. * In the case of cRGD-MID-AC, the CBE for normal brain tissue was unknown. Therefore, the boxes are kept blank.

## Data Availability

The datasets analyzed in the current study are available from the corresponding author upon reasonable request. The JMP Pro version 15.1.0. software (SAS, Cary, NC, USA) was used for the statistical analysis.

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
