# Peer review of "Improved Boron Neutron Capture Therapy Using Integrin αvβ3-Targeted Long-Retention-Type Boron Carrier in a F98 Rat Glioma Model"

_biology, 2023, doi:10.3390/biology12030377_

Round 1

Reviewer 1 Report

This is an expertly written manuscript describing the biodistribution and therapeutic evaluation of a cRGD-targeted HSA construct bearing 10B-decaborane for BNCT in an orthotopic model of rat F98 glioma. The manuscript is very clear to follow for the most part and the data is presented according to the standards in the field, with healthy criticism towards the potential weaknesses in the experimental setup and without overselling the findings. Ultimately a modest improved therapeutic effect is seen after BNCT in animals treated with the novel boron carrier over ones treated with BPA fructose complex. There are a few points the authors could clarify in order to further improve the manuscript. These are detailed below.
1) section 2.1. Here the authors could explain better how was the selectivity with Cys over Lys residues achieved for different conjugations and give the detailed synthetic conditions. It is not properly described in reference 29 either where the strategy was originally reported.  Also a schematic of the structure would be very helpful so the reader wouldn't have to look it up in the references. The authors should justify the choice of immunogenic HSA as the carrier over rat serum albumin.

2) Section 3.5 There's a typo in 'estimation'

3) page 12, line 471 onwards: There are cellular assays where the membrane-bound fraction of the compound can be isolated. Were these considered? Also I think the internalization of alphavbetaintegrin is rather well studies, so a reference to the rate of internalization and mechanism from literature could be made.

4) page 12, line 493: 'Prospective' is not the correct choice of the word here I think.

5) page 12, line 498: note that the mass number of the fluorine-18 isotope should be in superscript  

Author Response

Comments from reviewer 1

  • section 2.1. Here the authors could explain better how was the selectivity with Cys over Lys residues achieved for different conjugations and give the detailed synthetic conditions. It is not properly described in reference 29 either where the strategy was originally reported. Also a schematic of the structure would be very helpful so the reader wouldn’t have to look it up in the references. The authors should justify the choice of immunogenic HSA as the carrier over rat serum albumin.

Response to Reviewer 1

 Thank you for pointing this out. Regarding the binding selectivity of human serum albumin to lysine residues, there have been reports in the past, and we have added the following reference as a reference. We have also added a schema showing the structural formula and synthetic route of cRGD-MID-AC as Figure A1.

Reference: Ishii, S.; Sato, S-I.; Asami, H.; Hasegawa, T.; Kohno, J-Y.; Nakamura, H. Design of S-S bond containing maleimide-conjugated closo-dodecaborate (SSMID): identification of unique modification sites on albumin and investigation of intracellular uptake. Org Biomol Chem. 2019, 17(22), 5496-5499. https://doi.org/10.1039/c9ob00584f.

As you indicated, human serum albumin was used in this experiment, not rat serum albumin. We need to consider the effect of immunogenity. The following text is added in the Discussion section.

When considering this albumin-containing drug as a pre-clinical study, the results of this study require to be interpreted carefully, taking into account the effects of immunogenicity in a rat brain tumor model.

Comments from reviewer 1

  • Section 3.5 There’s a typo in ‘estimation’

Response to Reviewer 1

Thank you for pointing this out. It has been corrected to ‘estimation’.

Comments from reviewer 1

  • Page 12, line 471 onwards: there are cellular assays where the membrane-bound fraction of the compound can be isolated. Were these considered? Also I think the internalization of alphavbeta3 integrin is rather well studies, so a reference to the rate of internalization and mechanism from literature could be made.

Response to Reviewer 1

Thank you for pointing this out. We have not been able to perform the cellular assay you have suggested. This point is the limitation of this study. We consider that in the future we will plan to explore in the areas. In addition, descriptions of integrin internalization rates and mechanisms of uptake have been added to the text based on references.

Referrences:

De Deyne, P.G.; O'Neill, A.; Resneck, W.G.; Dmytrenko, G.M.; Pumplin, D.W.; Bloch, R.J. The vitronectin receptor associates with clathrin-coated membrane domains via the cytoplasmic domain of its beta5 subunit. J Cell Sci. 1998, 111 ( Pt18), 2729-40. https://doi.org/10.1242/jcs.111.18.2729

Alam, M.R.; Dixit, V.; Kang, H.; Li, Z.B.; Chen, X.; Trejo, J.; Fisher, M.; Juliano, R.L. Intracellular delivery of an anionic antisense oligonucleotide via receptor-mediated endocytosis. Nucleic Acids Res. 2008, 36(8):2764-76. https://doi.org/10.1093/nar/gkn115

Schraa, A.J.; Kok, R.J.; Berendsen, A.D.; Moorlag, H.E.; Bos, E.J.; Meijer, D.K.; de Leij, L.F.; Molema, G. Endothelial cells internalize and degrade RGD-modified proteins developed for tumor vasculature targeting. J Control Release. 2002 Oct 4;83(2):241-51. https://doi.org/10.1016/s0168-3659(02)00206-7

Sancey, L.; Lucie, S.; Garanger, E.; Elisabeth, G.; Foillard, S.; Stéphanie, F.; Schoehn, G.; Guy, S.; Hurbin, A.; Amandine, H.; Albiges-Rizo, C.; Corinne, A.R.; Boturyn, D.; Didier, B.; Souchier, C.; Catherine, S.; Grichine, A.; Alexeï, G.; Dumy, P.; Pascal, D.; Coll, J.L.; Jean-Luc, C. Clustering and internalization of integrin alphavbeta3 with a tetrameric RGD-synthetic peptide. Mol Ther. 2009, 17(5), 837-43. https://doi.org/10.1038/mt.2009.29

The Added text:

Although the interaction between RGD and integrins has long been known, the mechanism of internalization by binding of RGD peptides to integrins has not yet been cleared. However, it has been reported that at least the endocytosis of integrin αvβ3 is mediated via clathrin-dependent endocytosis or uncoated vesicles [49,50]. Schraa et al. showed that internalization of monomeric RGD ligands is independent of their αvβ3 receptor and occurs via a liquid-phase endocytic pathway. In contrast, multimeric RGD molecules are co-internalized with their receptor, with evidence supporting the aggregation and clustering of integrins [51]. More recently, Sancey et al. reported a peptide-like scaffold with four cRGD motifs, called RAFT-RGD that target integrin αvβ3 and promote integrin cluster formation. They have further demonstrated that the addition of 1 μmol/l of this molecule (RAFT-RGD) increases the internalization of αvβ3 via clathrin-coated vesicles by 79% [52]. Our study did not examine the internalization rate of Integrin αvβ3 or the extent to which our DDS applied with cRGD contributed to the internalization of Integrin αvβ3. However, we will explore these in future studies to better understand and improve the potential of DDS to specifically deliver and retain boron in target cells.

Comments from reviewer 1

  • Page 12, line 493: ‘Prospective’ is not the correct choice of the word here I think.

Response to Reviewer 1

Thank you for pointing this out. The choice of the word ‘ Prospective’ was not good. It has been corrected to ‘ Perspective’.

Comments from reviewer 1

  • Page 12, line 498: note that the mass number of the fluorine-18 isotope should be in superscript.

Response to Reviewer 1

Thank you for pointing this out. ‘18F’ has been corrected to ‘18F’. There was another same mistake, which has been corrected.

Message to Reviewer 1

The corrections in Materials and Methods part are at the request of MDPI's assistant editor. The content of the text remains unchanged.

Reviewer 2 Report

The manuscript describes the efficacy of BNCT with cRGD-MID-AC using a glioma-bearing rat brain tumor model. Nakamura and coworkers previously developed the cRGD-MID-AC, which targets integrin alpha-v beta-3, as a novel boron carrier for BNCT (ref. 29) and demonstrated its efficacy using the colon 26 tumor-bearing mice. In this paper, authors have successfully expanded its utility into treatment of glioma. Although the cellular uptake of cRGD-MID-AC in F98, C6 glioma, and 9L gliosarcoma cells was lower than that of BPA, the boron carrier brought significant BNCT effect and long-term survivors in the in vivo irradiation experiments. Since MID-AC has a great potential as a boron carrier for BNCT having a distinct accumulation mechanism, the reviewer will recommend this manuscript to be published in Biology after addressing the following issue. 

  Regarding the cellular uptake mechanism of cRGD-MID-AC, are there any experimental results that support the uptake via integrin alpha-v beta-3? In the previous report (ref.29), expression of integrin alpha-v beta-3 in various tumor cells was analyzed, but more direct evidence, such as competitive inhibition, have not been given do far.

Author Response

Comment from Reviewer 2

  • Regarding the cellular uptake mechanism of cRGD-MID-AC, are there any experimental results that support the uptake via integrin alpha-v beta-3? In the previous report (ref.29), expression of integrin alpha-v beta-3 in various tumor cells was analyzed, but more direct evidence, such as competitive inhibition, have not been given do far.

Response to Reviewer 2

Thank you for pointing this out. We have not been able to perform experiments to prove integrin-mediated uptake by competitive inhibition, including our previous studies. However, the mechanism of integrin-mediated cellular uptake of RGD peptides and their internalization has been reported, and we have cited the following references and added text. In the future, we also plan to conduct experiments to prove integrin-mediated cellular uptake of cRGD-MID-AC.

Referrences:

De Deyne, P.G.; O'Neill, A.; Resneck, W.G.; Dmytrenko, G.M.; Pumplin, D.W.; Bloch, R.J. The vitronectin receptor associates with clathrin-coated membrane domains via the cytoplasmic domain of its beta5 subunit. J Cell Sci. 1998, 111 ( Pt18), 2729-40. https://doi.org/10.1242/jcs.111.18.2729

Alam, M.R.; Dixit, V.; Kang, H.; Li, Z.B.; Chen, X.; Trejo, J.; Fisher, M.; Juliano, R.L. Intracellular delivery of an anionic antisense oligonucleotide via receptor-mediated endocytosis. Nucleic Acids Res. 2008, 36(8):2764-76. https://doi.org/10.1093/nar/gkn115

Schraa, A.J.; Kok, R.J.; Berendsen, A.D.; Moorlag, H.E.; Bos, E.J.; Meijer, D.K.; de Leij, L.F.; Molema, G. Endothelial cells internalize and degrade RGD-modified proteins developed for tumor vasculature targeting. J Control Release. 2002 Oct 4;83(2):241-51. https://doi.org/10.1016/s0168-3659(02)00206-7

Sancey, L.; Lucie, S.; Garanger, E.; Elisabeth, G.; Foillard, S.; Stéphanie, F.; Schoehn, G.; Guy, S.; Hurbin, A.; Amandine, H.; Albiges-Rizo, C.; Corinne, A.R.; Boturyn, D.; Didier, B.; Souchier, C.; Catherine, S.; Grichine, A.; Alexeï, G.; Dumy, P.; Pascal, D.; Coll, J.L.; Jean-Luc, C. Clustering and internalization of integrin alphavbeta3 with a tetrameric RGD-synthetic peptide. Mol Ther. 2009, 17(5), 837-43. https://doi.org/10.1038/mt.2009.29

The Added text:

Although the interaction between RGD and integrins has long been known, the mechanism of internalization by binding of RGD peptides to integrins has not yet been cleared. However, it has been reported that at least the endocytosis of integrin αvβ3 is mediated via clathrin-dependent endocytosis or uncoated vesicles [49,50]. Schraa et al. showed that internalization of monomeric RGD ligands is independent of their αvβ3 receptor and occurs via a liquid-phase endocytic pathway. In contrast, multimeric RGD molecules are co-internalized with their receptor, with evidence supporting the aggregation and clustering of integrins [51]. More recently, Sancey et al. reported a peptide-like scaffold with four cRGD motifs, called RAFT-RGD that target integrin αvβ3 and promote integrin cluster formation. They have further demonstrated that the addition of 1 μmol/l of this molecule (RAFT-RGD) increases the internalization of αvβ3 via clathrin-coated vesicles by 79% [52]. Our study did not examine the internalization rate of Integrin αvβ3 or the extent to which our DDS applied with cRGD contributed to the internalization of Integrin αvβ3. However, we will explore these in future studies to better understand and improve the potential of DDS to specifically deliver and retain boron in target cells.

Message to Reviewer 2

The corrections in Materials and Methods part are at the request of MDPI's assistant editor. The content of the text remains unchanged.
